# OpenReview forum: "Mitigating Over-Refusal in Adversarial Tuning via Subspace-guided Sample Selection"
_ICLR.cc/2026/Conference — Submitted to ICLR 2026_

### Official Review · Reviewer_VSnu · 2025-10-30

**Soundness:** 3
**Presentation:** 3
**Contribution:** 3
**Rating:** 8
**Confidence:** 3

**Summary:**

This paper addresses over-refusal in LLM adversarial tuning, which degrades utility. It proposes SAT, a subspace-guided soft sample selection framework. Key results: It reduces over-refusal by over 22%, maintains ASR below 2.8% across 5 attacks, and preserves utility (e.g., AlpacaEval winrate 54.61% vs SafeLoRA’s 47.37%).

**Strengths:**

1. Precise subspace decomposition: It splits model activation space into robustness and over-refusal subspaces, using gradient projection to quantify sample impact (e.g., projecting candidate gradients to select samples with strong robustness and weak over-refusal influence), enabling targeted tuning.
2. Extensive baseline comparisons: It compares 6 SOTA defenses (PPL, Self-Reminder, etc.) across 3 models (Vicuna, Llama2, Dolphin), showing SAT’s superiority (e.g., Vicuna’s SAT reduces ASR to 2.8% vs Self-Examination’s 10%).
 3. Utility preservation: Unlike Goal Priority (which harms GSM8K accuracy), SAT maintains or improves utility (e.g., Llama2’s GSM8K accuracy 53.20% vs SafeLoRA’s 52.20%).

**Weaknesses:**

1. Limited sample generation sources: It relies on GCG to generate candidate adversarial samples; other attack methods (e.g., AutoDAN, TAP) are untested—using diverse attack-generated samples could enhance sample selection generality.
2. Lack of long-term evaluation: It does not test over-refusal drift after extended fine-tuning (e.g., 10+ epochs); evaluating durability would confirm long-term effectiveness.
3. Narrow domain coverage: It only tests semantic QA and math reasoning; high-stakes domains (e.g., healthcare, finance) are unexamined—adding domain-specific tests would improve real-world relevance.

**Questions:**

Please refer to the weaknesses above.

---

> ### Author Response · Authors · 2025-11-24
>
> We sincerely thank you for your careful evaluation of our work and for the valuable feedback provided. Herein, we have thoroughly considered and verified each comment, and we provide detailed responses in the following sections.
>
> **1. W 1: limited sample generation sources**
>
> Thank you for your valuable comment. Indeed, candidate samples generated by different types of attacks may have varying impacts on the model’s defense performance. To further investigate this, we categorize attacks into black box and white box settings: GCG, used in our main experiments, is a white box attack, while in the comparative experiments we introduce the black box attack PAIR, as well as a hybrid setting that combines both. The corresponding experimental results are shown in the figure below. Experimental results indicate that fine-tuning samples derived from a specific attack type yield superior defensive
> performance against that same attack type. In contrast, samples generated via hybrid attacks exhibit no significant bias toward specific attack vectors; however, they demonstrate effective defense capabilities against both black-box and white-box attacks. We report the detailed results in Table 8 of Appendix 4 in the revised version of the paper.
>
> | Model | Defense | AdvBench ↓ | GCG | AutoDAN | PAIR | DeepInception | TAP | Average ↓ |
> | :--- | :--- | :---: | :---: | :---: | :---: | :---: | :---: | :---: |
> | **Vicuna-7B** | SAT(black-box) | 0% ± 0.0 | 0% ± 0.0 | 11% ± 1.2 | 0% ± 0.0 | 3% ± 1.0 | 1% ± 0.2 | 3.0% |
> | | SAT(white-box) | 0% ± 0.0 | 0% ± 0.0 | 12% ± 1.2 | 0% ± 0.0 | 0% ± 0.0 | 2% ± 0.4 | 2.8% |
> | | SAT(MIX) | 0% ± 0.0 | 0% ± 0.0 | 10% ± 1.1 | 1% ± 0.1 | 2% ± 0.1 | 3% ± 0.4 | 3.2% |
> | **Llama2-7B** | SAT(black-box) | 0% ± 0.0 | 1% ± 0.0 | 5% ± 1.4 | 3% ± 0.2 | 2% ± 0.0 | 1% ± 0.1 | 2.4% |
> | | SAT(white-box) | 0% ± 0.0 | 1% ± 0.2 | 0% ± 0.0 | 4% ± 0.3 | 0% ± 0.0 | 1% ± 1.0 | 1.2% |
> | | SAT(MIX) | 0% ± 0.0 | 0% ± 0.0 | 3% ± 1.2 | 4% ± 0.6 | 1% ± 0.1 | 2% ± 0.6 | 2.0% |
> | **Dolphin-7B** | SAT(black-box) | 1% ± 0.2 | 0% ± 0.0 | 7% ± 1.0 | 3% ± 0.4 | 2% ± 0.2 | 1% ± 0.7 | 2.6% |
> | | SAT(white-box) | 0% ± 0.0 | 0% ± 0.0 | 7% ± 1.1 | 2% ± 0.0 | 4% ± 0.2 | 1% ± 0.0 | 2.8% |
> | | SAT(MIX) | 0% ± 0.0 | 0% ± 0.0 | 9% ± 1.3 | 2% ± 0.4 | 4% ± 0.3 | 1% ± 0.2 | 3.2% |
> | **Qwen-7B** | SAT(black-box) | 0% ± 0.0 | 4% ± 0.7 | 7% ± 1.4 | 0% ± 0.0 | 8% ± 0.5 | 8% ± 0.6 | 5.4% |
> | | SAT(white-box) | 0% ± 0.0 | 3% ± 1.0 | 9% ± 1.1 | 0% ± 0.0 | 6% ± 0.4 | 9% ± 1.4 | 5.4% |
> | | SAT(MIX) | 0% ± 0.0 | 3% ± 0.2 | 9% ± 1.7 | 0% ± 0.0 | 6% ± 0.7 | 6% ± 0.6 | 4.8% |
>
> **2. W 2: Lack of long-term evaluation**
> We thank the reviewer for the insightful comment regarding the training duration. We would like to clarify that our decision to stop training at the current epoch is grounded in the theoretical framework of Low-Rank Adaptation (LoRA) and the Intrinsic Dimensionality of LLMs.
>
> a. **Rapid Convergence of LoRA :** According to the original LoRA paper [1], LoRA operates by optimizing rank-decomposition matrices, which significantly reduces the trainable parameter space. Unlike Full-Parameter Fine-Tuning that requires navigating a vast optimization landscape, LoRA focuses on the most salient features.
>
> b. **Risk of Overfitting :**
> Research by [2] demonstrates that pre-trained language models have a low "intrinsic dimension." This implies that the model can adapt to new tasks by modifying parameters within a very small subspace. As noted in recent PEFT literature, excessive training on a low-rank adapter when the loss is already minimal leads to overfitting to the training noise rather than learning generalizable features.
>
> Therefore, the current stopping point was deliberately chosen to maximize the trade-off between task adaptation and generalization, adhering to the best practices established in recent literature.
>
> [1]. Hu, Edward J., et al. "Lora: Low-rank adaptation of large language models." ICLR 1.2 (2022): 3.
>
> [2]. Aghajanyan, Armen, Sonal Gupta, and Luke Zettlemoyer. "Intrinsic dimensionality explains the effectiveness of language model fine-tuning." Proceedings of the 59th annual meeting of the association for computational linguistics and the 11th international joint conference on natural language processing (volume 1: long papers). 2021.
>
> **3. W 3: Narrow domain coverage**
>
> We greatly appreciate this insightful suggestion. We agree that evaluating our method in high-stakes domains like healthcare and finance is crucial for demonstrating its real-world robustness. In this work, our primary objective was to validate the fundamental effectiveness of our proposed approach. Therefore, we selected Semantic QA and Math Reasoning as our evaluation benchmarks because they are widely recognized standards for assessing a model's core capabilities in logical reasoning and language understanding. We will incorporate your suggestion regarding healthcare and finance in our future work as the next step for real-world deployment.

---

### Official Review · Reviewer_dshb · 2025-10-30

**Soundness:** 2
**Presentation:** 3
**Contribution:** 2
**Rating:** 4
**Confidence:** 4

**Summary:**

This paper tackles a common side-effect of adversarial fine-tuning for LLM safety: over-refusal of benign queries. The authors propose Soft Adversarial Tuning (SAT), a data-selection framework that: (i) constructs two behavior subspaces from hidden activations, one for jailbreak robustness and one for over-refusal avoidance, using difference-in-means “steering” vectors; (ii) projects per-sample gradients onto these subspaces to estimate how each candidate training example would shift behavior; and (iii) scores and selects “soft” samples that push strongly along the robustness direction while minimally affecting the over-refusal direction. The selected set is then used for LoRA fine-tuning. Evaluated on Vicuna-7B, Llama-2-7B-Chat, and Dolphin-7B against five jailbreak attacks, SAT is reported to lower attack success rates and reduce refusal on safe inputs relative to several baselines, with an ablation indicating the explicit over-refusal penalty in the scoring term is important.

**Strengths:**

1.	The paper centers on over-refusal on LLM, then proposes a specific mechanism (dual subspaces, gradient projections, sample scoring) rather than an open-ended recipe. The subspace construction via contrastive positive/negative pairs and difference-in-means vectors is well-defined and easy to implement.
2.	SAT is a pre-fine-tuning data curation step that could be bolted onto diverse PEFT schemes; the actual model update uses standard LoRA, keeping the method pragmatic for practitioners.
3.	On three 7B chat models and five representative jailbreaks, SAT often shows lower ASR and lower harmful scores than baselines such as Self-Examination, ICD, Retokenization, and Paraphrase; safe-set refusal is also reduced on XSTest. The tables make these cross-method comparisons explicit.

**Weaknesses:**

1.	The approach assumes a linear correspondence between activation-space projections at a single layer and subsequent parameter-space updates, then uses one-dimensional directions per behavior to rank samples. There is no sensitivity analysis across layers, pooling choices, or multi-dimensional subspaces, and no theoretical or empirical justification that a single direction captures over-refusal vs robustness without leakage between them. At minimum, a study varying layer index, using multi-basis PCA/LDA subspaces, and checking orthogonality would be needed.
2.	Since SAT is a sample selection method, it should be compared against strong selection baselines: e.g., gradient-norm filtering, loss-based filtering, influence-function/DShapley-style data valuation, or even simple intermediate-iteration only heuristics. Current baselines are mostly defense mechanisms at inference or full training, not data curation approaches, so the incremental value of subspace-guided selection is unclear.
3.	The pipeline generates 500 optimized candidates per seed over 400 seeds, then projects gradients for scoring yet finally keeps only 500 samples. There is no accounting of attack-generation time, gradient-projection overhead, or overall wall-clock vs baselines. For a method pitched as “efficient pre-selection,” this omission is important.

**Questions:**

1.	Table 1 mixes ASR and harmful scores per cell but has a few oddities/typos (e.g., ICA, duplicated averages formatting) and the narrative highlights dramatic gains without confidence intervals, seed variability, or per-category breakdowns. The claim reduce over-refusal by more than 22% while keeping ASR below 2.8% would benefit from statistical tests across runs and categories.
2.	One figure panel includes an error message (index out of bounds / lda (Failed)), which undermines polish and raises questions about the robustness of the dimensionality-reduction diagnostics. Implementation details crucial to reproducing subspaces are not fully specified.

---

> ### Author Response · Authors · 2025-11-24
>
> Thank you very much for your insightful suggestions you shared. We have carefully gone through all of your comments one by one, verified the relevant issues, and organized our point-by-point responses in the following sections.
>
> **1. W 1:**
>
> a. **Layer-wise Sensitivity Analysis:** We have included a visualization in Figure14 of Appendix.4 in the revision, which demonstrates how the separability of positive and negative samples changes across layers from shallow to deep. The representations of positive and negative samples are entangled in early layers but become progressively separable in deeper layers. This indicates that higher layers exhibit more refined semantic discrimination compared to shallower layers, making them better suited for subspace partitioning.
>
> b. **Justification for 1D Direction:** The selection of the difference-in-means vector, $v_k$, is motivated by its objective to maximize class separability, a goal that directly corresponds to the task of distinguishing between the two behaviors. Actually, recent studies[1][2][3] increasingly reveal that within the activation spaces of LLMs, representations of high-level abstract concepts like safety and honesty show remarkable linear structure. This means their essential characteristics can be effectively approximated by a single direction vector. This discovery has given rise to a powerful and increasingly prevalent approach: deriving this vector by computing the difference-in-means between the activations of positive and negative samples.
>
> [1]. Zou, Andy, et al. "Representation engineering: A top-down approach to ai transparency, 2023." URL https://arxiv.org/abs/2310.01405 97 (2022).
>
> [2]. Turner, Alex, et al. "Steering gpt-2-xl by adding an activation vector." AI Alignment Forum. 2023.
>
> [3]. Perez, Ethan, et al. "Red teaming language models with language models." arXiv preprint arXiv:2202.03286 (2022).
>
> **2. W 2: SAT should be compared against strong selection baselines**
>
> We conducted a comparative evaluation of three data selection methods: random sampling, loss-based optimized selection, and manual construction of benign samples. Key experimental results are presented below, while the complete experimental data and a full analysis can be found in Table 7 of Appendix.4 of the revised manuscript
>
> | Model | Defense | AdvBench $\downarrow$ | GCG | AutoDAN | PAIR | DeepInception | TAP | Average $\downarrow$ |
> | :--- | :--- | :---: | :---: | :---: | :---: | :---: | :---: | :---: |
> | **Vicuna-7B** | Random-based | 2% ± 0.8 | 17% ± 1.2 | 57% ± 2.3 | 62% ± 2.0 | 49% ± 1.9 | 63% ± 1.4 | 49.6% |
> | **Vicuna-7B** | Human construction | 7% ± 0.3 | 20% ± 1.0 | 72% ± 1.2 | 64% ± 1.5 | 70% ± 1.6 | 52% ± 0.4 | 55.6% |
> | **Vicuna-7B** | loss-based | 1% ± 0.0 | 11% ± 0.7 | 19% ± 1.5 | 9% ± 0.9 | 11% ± 0.4 | 14% ± 0.4 | 12.8% |
> | **Vicuna-7B** | SAT | 0% ± 0.0 | 0% ± 0.0 | 12% ± 1.2 | 0% ± 0.0 | 0% ± 0.0 | 2% ± 0.4 | 2.8% |
> | **Llama2-7B** | Random-based | 5% ± 0.1 | 12% ± 0.4 | 44% ± 0.7 | 57% ± 0.3 | 60% ± 0.8 | 71% ± 1.8 | 48.8% |
> | **Llama2-7B** | Human construction | 3% ± 0.0 | 14% ± 0.3 | 38% ± 0.6 | 42% ± 0.7 | 41% ± 0.4 | 53% ± 1.6 | 37.6% |
> | **Llama2-7B** | loss-based | 0% ± 0.0 | 10% ± 0.1 | 21% ± 0.5 | 26% ± 0.3 | 19% ± 0.4 | 22% ± 1.3 | 19.6% |
> | **Llama2-7B** | SAT | 0% ± 0.0 | 1% ± 0.2 | 0% ± 0.0 | 4% ± 0.3 | 0% ± 0.0 | 1% ± 1.0 | 1.2% |
> | **Dolphin-7B** | Random-based | 4% ± 0.1 | 19% ± 1.1 | 47% ± 1.6 | 58% ± 0.9 | 64% ± 0.7 | 61% ± 0.6 | 49.8% |
> | **Dolphin-7B** | Human construction | 3% ± 0.4 | 13% ± 0.5 | 32% ± 1.1 | 34% ± 0.9 | 29% ± 0.2 | 31% ± 0.7 | 27.8% |
> | **Dolphin-7B** | loss-based | 1% ± 0.0 | 10% ± 0.0 | 27% ± 1.4 | 21% ± 0.3 | 14% ± 0.2 | 19% ± 0.0 | 18.2% |
> | **Dolphin-7B** | SAT | 0% ± 0.0 | 0% ± 0.0 | 7% ± 1.1 | 2% ± 0.0 | 4% ± 0.2 | 1% ± 0.0 | 2.8% |
> | **Qwen-7B** | Random-based | 2% ± 0.0 | 15% ± 0.4 | 59% ± 1.6 | 63% ± 1.0 | 45% ± 0.4 | 59% ± 1.7 | 48.2% |
> | **Qwen-7B** | Human construction | 1% ± 0.0 | 13% ± 0.2 | 53% ± 1.2 | 58% ± 0.6 | 38% ± 0.2 | 43% ± 1.1 | 41.0% |
> | **Qwen-7B** | loss-based | 0% ± 0.0 | 12% ± 1.0 | 22% ± 0.1 | 17% ± 0.3 | 16% ± 0.0 | 21% ± 1.3 | 17.6% |
> | **Qwen-7B** | SAT | 0% ± 0.0 | 3% ± 1.0 | 9% ± 1.1 | 0% ± 0.0 | 6% ± 0.4 | 9% ± 1.4 | 5.4% |

---

> > ### Author Response · Authors · 2025-11-24
> >
> > **3. W3: The paper claims it is an efficient method, but there is no accounting of attack-generation time**
> >
> > Thank you for your comment. We apologize that our explanation of the term "efficient" was not sufficiently clear and may have led to a misunderstanding. To clarify, the "efficiency" we refer to does not mean that our selection process itself is faster in terms of wall-clock time than methods like random sampling. Instead, our claim of efficiency is based on the following two points:
> >
> > a. **Efficiency within the PEFT Context:** Our work is situated within the paradigm of Parameter-Efficient Fine-Tuning (PEFT). A significant part of the overall efficiency is inherited from this paradigm, which avoids full-parameter fine-tuning. Our data pre-selection method serves as an additional optimization layer, designed to maximize the performance of these already-efficient tuning methods.
> >
> > b. **Efficiency through Predictive Power (Our Core Contribution):** The primary efficiency gain of our method lies in its ability to act as a "proxy" for final model performance. By investing a relatively small amount of computational resources upfront in our selection process, we can predict which data subsets will yield better results *before* committing to expensive, resource-intensive full fine-tuning runs. A single full fine-tuning process can consume hours or even days of GPU time. Our method helps researchers avoid wasting these valuable resources on suboptimal data, thereby making the entire model development lifecycle significantly more efficient.
> >
> > **4. Q1: ASR without confidence intervals**
> > Our experimental results are based on the average over multiple role‑playing runs. To increase the diversity of the model outputs, we also set the sampling method to top‑p sampling. In the revised version of the paper, we have added the standard deviation of the experimental results to report stability. A portion of the results is shown in the figure below; please refer to the revised paper for the complete experimental results.
> >
> > | Model | Defense | AdvBench $\downarrow$ | GCG | AutoDAN | PAIR | DeepInception | TAP | Average $\downarrow$ |
> > | :--- | :--- | :---: | :---: | :---: | :---: | :---: | :---: | :---: |
> > | **Vicuna-7B** | No Defense | 8% ± 2.7 | 100% ± 0.0 | 98% ± 1.4 | 90% ± 2.0 | 81% ± 1.9 | 63% ± 1.8 | 86.4% |
> > | **Vicuna-7B** | PPL | 7% ± 2.6 | 0% ± 0.0 | 88% ± 2.2 | 88% ± 1.2 | 86% ± 1.5 | 59% ± 1.9 | 64.2% |
> > | **Vicuna-7B** | Self-Reminder | 2% ± 1.4 | 48% ± 2.0 | 68% ± 1.7 | 46% ± 2.2 | 62% ± 1.9 | 57% ± 1.5 | 56.2% |
> > | **Vicuna-7B** | Retokenization | 25% ± 1.3 | 42% ± 0.9 | 84% ± 1.7 | 82% ± 0.8 | 79% ± 1.1 | 38% ± 0.9 | 65% |
> > | **Vicuna-7B** | ICD | 0% ± 0.0 | 68% ± 1.7 | 80% ± 2.0 | 60% ± 2.1 | 46% ± 1.0 | 12% ± 1.2 | 53.2% |
> > | **Vicuna-7B** | Self-Examination | 0% ± 0.0 | 21% ± 1.1 | 12% ± 1.6 | 9% ± 1.3 | 6% ± 0.4 | 3% ± 0.2 | 10% |
> > | **Vicuna-7B** | Paraphrase | 10% ± 1.0 | 25% ± 1.3 | 70% ± 1.6 | 27% ± 1.4 | 51% ± 1.0 | 26% ± 1.4 | 39.8% |
> > | **Vicuna-7B** | **SAT(ours)** | **0% ± 0.0** | **0% ± 0.0** | **12% ± 1.2** | **0% ± 0.0** | **0% ± 0.0** | **2% ± 0.4** | **2.8%** |
> >
> > **5. Q2: Error message (index out of bounds / lda (Failed)) in paper**
> >
> > We have carefully checked the paper and did not find the error message (“index out of bounds” / “lda (Failed)”) mentioned by the reviewer.  If the issue persists, we would be very grateful if the reviewer could kindly indicate the specific location so that we can correct it promptly.

---

### Official Review · Reviewer_mb6v · 2025-10-31

**Soundness:** 3
**Presentation:** 2
**Contribution:** 3
**Rating:** 4
**Confidence:** 4

**Summary:**

This paper addresses the "over-refusal" problem in LLMs. This is when models reject benign queries after adversarial tuning. The authors propose the Soft Adversarial Tuning (SAT) framework. SAT uses representation engineering to define two subspaces: a "robustness subspace" and an "over-refusal subspace". It then uses gradient projections to select "soft samples". These samples have a strong influence on robustness but a minimal effect on over-refusal. The main contribution is this automatic sample selection mechanism.

**Strengths:**

1. The paper is well-written and easy to understand. The figures are also clear and well-done.
2. The paper's method is highly novel. The experimental results also show that the proposed method is effective.

**Weaknesses:**

1. The quotation marks for "soft samples" in the abstract are not correct.

2. The paper uses mean-pooled hidden states from layer $l$. Many methods use the hidden state of the last token. The authors should explain why they chose mean-pooling and discuss how this choice impacts the results.

3. Please provide a detailed explanation for the rationale behind Equation (8). Specifically, why is the absolute value of $p_2$ used? The paper describes $v_2$ as the direction vector pointing from the over-refusal space to the normal response space. Does this imply that more robust samples also tend to cause over-refusal, thereby resulting in a negative $p_2$ value?

4. Equation (5) is unclear. Please specify what the gradient is taken with respect to.

5. The paper has small formatting problems. For example, lines 316 and 297 are missing spaces after the colons. Many citation formats are also incorrect.

6. A formatting error: The caption for a table should be placed above it.

7. It would be beneficial to add results on the latest Qwen models.

**Questions:**

These are all in the 'Weaknesses' section above.

---

> ### Author Response · Authors · 2025-11-24
>
> We sincerely thank you for your careful evaluation of our work and for the valuable feedback provided. Herein, we have thoroughly considered and verified each comment, and we provide detailed responses in the following sections.
>
> **1. W 1&5&6:**
>
> Thank you for your suggestions regarding the clarity and presentation of the paper. We have carefully re-checked the formatting throughout the manuscript and have incorporated the corresponding revisions in the updated version.
>
> * We have revised the formatting of quotation marks in the Abstract.
> * We have added spaces after colons.
> * We have moved all table captions to be placed above the tables.
> * We have corrected the formatting of all reference citations.
> * We have thoroughly checked the grammar and formatting throughout the manuscript.
>
>
> **2. W 2: Why do the authors use mean pooling over the hidden states to obtain representations instead of using the last token?**
>
> We thank the reviewer for this insightful question and the suggestion to explore the last token representation. Our initial choice of Mean Pooling was primarily motivated by the nature of our subspace partitioning task, which relies on semantic clustering. As demonstrated in the work of Reimers et al. [1], mean pooling is robust in extracting global semantic information, particularly for tasks such as clustering and retrieval. However, following the reviewer's suggestion, and acknowledging the computational efficiency of the last token approach, we conducted additional experiments using the last token representation. The results are compared below. We found that both methods achieved comparable overall performance, suggesting that the efficacy of SAT is relatively insensitive to the specific choice of pooling strategy. We appreciate this valuable suggestion, as it strengthens the robustness analysis of our method. We have added a discussion regarding this aspect in Table 3 of Section 4.3 in the revised manuscript.
>
> [1]. Reimers, Nils, and Iryna Gurevych. "Sentence-bert: Sentence embeddings using siamese bert-networks." *arXiv preprint arXiv:1908.10084* (2019).
> | Model | Defense | AdvBench $\downarrow$ | GCG | AutoDAN | PAIR | DeepInception | TAP | Average $\downarrow$ |
> | :--- | :--- | :---: | :---: | :---: | :---: | :---: | :---: | :---: |
> | **Vicuna-7B** | SAT(M) | 0% ± 0.0 | 0% ± 0.0 | 12% ± 1.2 | 0% ± 0.0 | 0% ± 0.0 | 2% ± 0.4 | 2.8% |
> | | SAT(L) | 0% ± 0.0 | 0% ± 0.0 | 10% ± 1.4 | 0% ± 0.0 | 2% ± 0.0 | 3% ± 0.6 | 3.0% |
> | **Llama2-7B** | SAT(M) | 0% ± 0.0 | 1% ± 0.2 | 0% ± 0.0 | 4% ± 0.3 | 0% ± 0.0 | 1% ± 1.0 | 1.2% |
> | | SAT(L) | 0% ± 0.0 | 0% ± 0.0 | 0% ± 0.0 | 5% ± 0.4 | 0% ± 0.0 | 0% ± 0.0 | 1.0% |
> | **Dolphin-7B** | SAT(M) | 0% ± 0.0 | 0% ± 0.0 | 7% ± 1.1 | 2% ± 0.0 | 4% ± 0.2 | 1% ± 0.0 | 2.8% |
> | | SAT(L) | 0% ± 0.0 | 0% ± 0.0 | 9% ± 1.4 | 2% ± 0.1 | 2% ± 0.2 | 0% ± 0.0 | 2.6% |
> | **Qwen-7B** | SAT(M) | 0% ± 0.0 | 3% ± 1.0 | 9% ± 1.1 | 0% ± 0.0 | 6% ± 0.4 | 9% ± 1.4 | 5.4% |
> | | SAT(L) | 0% ± 0.0 | 1% ± 1.3 | 11% ± 1.7 | 2% ± 0.2 | 6% ± 0.3 | 8% ± 1.0 | 5.6% |
>
> | over-refusal | vicuna-7B-SAT(M) | vicuna-7B-SAT(L) | llama-2-7B-SAT(M) | llama-2-7B-SAT(L) | Dolphin-7B-SAT(M) | Dolphin-7B-SAT(L) | Qwen-7B-SAT(M) | Qwen-7B-SAT(L) |
> | :--- | :---: | :---: | :---: | :---: | :---: | :---: | :---: | :---: |
> | safe $\downarrow$ | 19.6% ± 0.7 | 18.3% ± 0.6 | 20.1% ± 0.8 | 18.9% ± 1.0 | 16.2% ± 0.8 | 16.3% ± 0.4 | 19.3% ± 0.9 | 20.5% ± 0.9 |
> | unsafe $\uparrow$ | 90.0% ± 1.3 | 90.3% ± 1.1 | 92.8% ± 0.9 | 92.9% ± 0.5 | 91.5% ± 1.2 | 90.3% ± 1.4 | 91.6% ± 1.0 | 92.1% ± 0.8 |
>
> **3. W 3: Why is the absolute value used for p2 in the Equation (8)**
>
> We appreciate the reviewer's query regarding the constraint on $p_2$. Intuitively, penalizing only negative values seems sufficient, as a positive $p_2$ might suggest an improvement in general capabilities. However, we employ the absolute value term $|p_2|$ based on two critical considerations:
>
> a. The fine-tuning samples (malicious prompts + refusal response) inherently lack supervision signals for general knowledge. Therefore, a positive $p_2$ might reflect spurious correlations, such as shifts in vocabulary distribution or changes in writing style, meaning these samples could introduce instability. If we were to allow directional optimization, the optimizer may tend to select and fit these samples with noisy features, which leads to instability.
>
> b. Our primary objective is disentanglement. In other words, improving safety robustness while strictly preserving the model's original utility on harmless queries. The absolute value term serves as an orthogonality constraint, forcing the projection of selected samples onto the "over-refusal" dimension to be minimal. This ensures that the features of the selected adversarial fine-tuning samples are orthogonal to those of harmless queries

---

> > ### Author Response · Authors · 2025-11-24
> >
> > **4. W4: What the gradient is taken with respect to in the Equation (5)?**
> >
> > We thank the reviewer for pointing out this ambiguity.
> >
> > In Equation (5), the gradient $\mathbf{g}_j$ is taken with respect to the hidden states (activations) at the selected layer $l$, denoted as $f_l(\mathbf{q}_j)$, rather than the model parameters.
> >
> > We chose to differentiate with respect to activations to predict the potential influence of a single sample on the model's behavior, without performing a full and computationally expensive backpropagation and parameter update. This gradient $\mathbf{g}_j$ indicates the direction in which the activations should be adjusted to reduce the loss for that sample, which we term the "expected update direction," and use as the basis for subsequent sample scoring.
> >
> > **5. W5: Add results on the Qwen models**
> >
> > Partial experimental results for the Qwen2.5-7B-Instruct model are shown below. The complete results have been updated in Section 4.2 of the revised manuscript.
> >
> > | Model | Defense | AdvBench $\downarrow$ | GCG | AutoDAN | PAIR | DeepInception | TAP | Average $\downarrow$ |
> > | :--- | :--- | :---: | :---: | :---: | :---: | :---: | :---: | :---: |
> > | **Qwen-7B** | No Defense | 4% ± 0.3 | 62% ± 2.3 | 57% ± 1.7 | 71% ± 1.5 | 33% ± 2.0 | 53% ± 1.4 | 55.2% |
> > | **Qwen-7B** | PPL | 4% ± 0.0 | 4% ± 0.9 | 25% ± 0.3 | 60% ± 2.1 | 8% ± 0.1 | 59% ± 1.6 | 31.2% |
> > | **Qwen-7B** | Self-Reminder | 1% ± 1.0 | 23% ± 0.2 | 22% ± 3.0 | 32% ± 1.7 | 12% ± 0.2 | 12% ± 0.2 | 20.2% |
> > | **Qwen-7B** | Retokenization | 11% ± 0.6 | 41% ± 1.7 | 36% ± 1.3 | 32% ± 2.0 | 17% ± 0.4 | 21% ± 1.1 | 29.4% |
> > | **Qwen-7B** | ICD | 0% ± 0.0 | 18% ± 0.9 | 42% ± 1.1 | 29% ± 1.5 | 3% ± 0.0 | 17% ± 1.2 | 21.8% |
> > | **Qwen-7B** | Self-Examination | 1% ± 0.0 | 18% ± 0.8 | 16% ± 0.7 | 13% ± 0.4 | 16% ± 0.3 | 13% ± 0.0 | 15.2% |
> > | **Qwen-7B** | Paraphrase | 22% ± 0.0 | 20% ± 0.0 | 11% ± 1.1 | 29% ± 0.5 | 19% ± 0.1 | 26% ± 1.0 | 21.0% |
> > | **Qwen-7B** | **SAT(ours)** | **0% ± 0.0** | **3% ± 1.0** | **9% ± 1.1** | **0% ± 0.0** | **6% ± 0.4** | **9% ± 1.4** | **5.4%** |

---

### Meta-Review · Area_Chair_y1Na · 2026-01-07

**Summary:**

The paper proposes Soft Adversarial Tuning, a method to mitigate over-refusal in adversarial training. It uses representation engineering to decompose hidden states into "robustness" and "over-refusal" subspaces, then projects gradients of candidate samples onto these subspaces to select "soft samples" for fine-tuning. The paper received mixed reviews with one supporting the paper for its empirical performance and utility preservation. Others raised significant concerns regarding the theoretical validity of the subspace decomposition, the lack of appropriate data selection baselines, and the practical efficiency of the pipeline.

**Reviewer Concerns:**

Reviewer Concerns Addressed: The authors provided additional experiments on Qwen models and compared their method against simple loss-based and random selection baselines.

Outstanding:

- Soundness: The core premise relies on a strong assumption that complex behaviors like "refusal" and "robustness" can be captured by single, linear directions in the activation space. The reviewers noted this is a significant oversimplification without sufficient theoretical backing or sensitivity analysis across layers.

- Practical Efficiency: The proposed pipeline is computationally heavy. It requires generating thousands of adversarial samples using expensive attacks (like GCG) before the selection process even begins. As noted in the critique of the efficiency claims, this "pre-selection" overhead makes the method less practical compared to standard curation or heuristic filtering.

- Baseline Comparisons: While defense baselines were provided, the comparison against state-of-the-art data selection or alignment techniques (beyond simple loss/random) remains weak. The effectiveness of this complex geometric filtering over simpler active learning methods is not decisively proven.

**Reviewer Scores:**

currently 448, not likely to change

---

### Decision · Program_Chairs · 2026-01-26

Reject